# The Role of Poly(ADP-ribose) Polymerase 1 in Nuclear and Mitochondrial Base Excision Repair

**DOI:** 10.3390/biom13081195

**Published:** 2023-07-31

**Authors:** Geoffrey K. Herrmann, Y. Whitney Yin

**Affiliations:** 1Department of Biochemistry and Molecular Biology, University of Texas Medical Branch, Galveston, TX 77555, USA; gkherrma@utmb.edu; 2Sealy Center for Structural Biology, University of Texas Medical Branch, Galveston, TX 77555, USA

**Keywords:** PARP-1, nuclear and mitochondrial localization, DNA repair

## Abstract

Poly(ADP-ribose) (PAR) Polymerase 1 (PARP-1), also known as ADP-ribosyl transferase with diphtheria toxin homology 1 (ARTD-1), is a critical player in DNA damage repair, during which it catalyzes the ADP ribosylation of self and target enzymes. While the nuclear localization of PARP-1 has been well established, recent studies also suggest its mitochondrial localization. In this review, we summarize the differences between mitochondrial and nuclear Base Excision Repair (BER) pathways, the involvement of PARP-1 in mitochondrial and nuclear BER, and its functional interplay with other BER enzymes.

## 1. The Importance of Mitochondrial Genome Integrity

Human cells contain two distinct genomes, the nuclear genome, consisting of ~3.2 billion base pairs (bp) linear DNA [1], and the mitochondrial genome, consisting of ~16.6 Kbp circular DNA (mtDNA) [2]. Each mitochondrion contains multiple copies of mtDNA within the inner mitochondrial matrix. Many metabolic processes occur within the mitochondrial membrane, creating an oxidatively damaging environment and endangering the mtDNA. Maintaining genome integrity requires the frequent and efficient repair of DNA lesions, which occur via spontaneous and non-spontaneous mechanisms. There are an estimated 10^4^–10^5^ DNA lesions per cell per day [3,4], and the mitochondrial genome is subjected to higher levels of DNA damage than the nuclear genome [5]. Intriguingly, the oxidatively damaged form of guanine, 8-oxoG, is reported to be repaired more efficiently in mitochondria relative to the nucleus [6]. Based on this, it can be argued that different machinery is involved during mtBER. Two lines of evidence support this argument: 1) certain nuclear BER enzymes have not been observed in mitochondria; 2) mitochondria-specific repair enzymes have been observed to participate in mtBER [7,8].

The vast majority of nuclear DNA is considered non-coding, ~97%, in stark contrast to the mitochondrial genome, where only ~7% of the genome is non-coding [9]. Thus, in addition to being subjected to higher levels of DNA damage, there is a greater chance that the damaged DNA will occur in a coding region, which could impact the gene product. Indeed, a litany of diseases are linked to mutations in the mitochondrial genome [10]; improving our understanding of the enzymes involved in mitochondrial genome maintenance would provide valuable insight.

The mitochondrial genome encodes 37 gene products, with the majority encoding ribosomal components (22 tRNAs, 2 rRNAs) and 13 proteins that are subunits for the oxidative phosphorylation electron transport chain [11]. Interestingly, the mitochondrial genome does not encode any enzymes for repairing the mitochondrial genome. Instead, mitochondria rely on nuclear-encoded enzymes that must be transported into mitochondria. However, not all proteins involved in nuclear DNA repair are imported into mitochondria, and mitochondria do not have the capability to perform all types of DNA repair processes [8]. Notably, nucleotide excision repair has not been observed in mitochondria [12]. Mitochondrial DNA suffers high levels of oxidative damage [13], which is primarily repaired by base excision repair, and the exact enzymes involved in this process in mitochondria are debated.

## 2. An Enzymatic Overview of Poly(ADP-ribose) Polymerase 1

The human PARP superfamily of enzymes currently has 17 known members. PARPs are involved in a variety of cellular functions, but are most famous for their involvement in DNA repair [14]. The name “PARP” is inaccurate for most PARP family members as only four enzymes catalyze ADP-ribose into a polymeric chain [15], while twelve are mono-ADP-ribosyl transferases (MARTs), and one has no catalytic activity. The key feature that allows these 17 proteins to be grouped into a single family is a conserved ADP-ribosyl transferase (ART) fold, which is located in a C-terminal domain in the 16 members, except for PARP-4, which is near the N-terminus [14,16].

The DNA damage response PARPs, PARPs 1–3, are the most well-known. PARP-1 generates the majority of intracellular PARylation (~90%) [17]. PARP-1 and PARP-2 play important roles in embryonic development and a PARP-1/PARP-2 double knockout is lethal, causing death at gastrulation [18]. Contrarily, PARP-1/PARP-3 double knockout is not lethal in murine cells [19]. All three of the DNA damage response PARPs have a high degree of similarity at the C-terminus, and the major difference occurs at the N-terminus: PARP-1 contains three N-terminal Zinc finger domains (Zn1, Zn2, and Zn3) and a domain containing a BRCT fold (BRCT), while PARP-2 and PARP-3 have a singular N-terminal domain (NTD) [20]. The C-terminal domains of the three proteins are highly similar, containing a WGR domain (rich in Trp, Gly, and Arg residues) and a catalytic (CAT) domain comprised of a helical subdomain (HD) and an ART fold (Figure 1A). The mode of DNA binding differs between PARP-1 and PARP-2; while the three major domains (NTD, WGR, CAT) work collaboratively to bind DNA in PARP-2, they contribute little to the DNA binding affinity in PARP-1, and its primary binding energy is contributed by Zn1 and Zn2 (Figure 1B), which are absent in PARP-2. Individual PARP-2 domains showed lower binding affinity than the full-length (FL) protein, regardless of DNA construct, as in [21]. Contrarily, the PARP-1 Zn1-Zn2 fragment displays an affinity for DNA that is comparable to FL, while the C-terminal region (Zn3, BRCT, WGR, and CAT) was unable to bind DNA effectively [22]. The DNA-binding mechanism of PARP-3 has not been explored to the same extent as PARP-1 and PARP-2, but current studies seem to indicate PARP-3 would recognize DNA in a manner similar to PARP-2. For both PARP-2 and PARP-3, their N-terminal regions are not strictly required for DNA-dependent activation, and the WGR domain is essential for catalytic activation in the presence of DNA [20].

Extensive biochemical and structural studies have answered key questions regarding PARP-1′s catalytic activation. First, it was established that PARP-1 required DNA binding to become catalytically active. It was later determined that four of the six domains are strictly required for catalytic activation: Zn1, Zn3, WGR, and CAT [23]. Zn1 makes extensive contacts with the DNA backbone and forms interfaces with the other three essential domains (Figure 1B). Ultimately, an internal auto-inhibitory helical subdomain (HD), which resides within the larger CAT domain, destabilizes, allowing sufficient space for NAD+ binding to the active site [23]. Hydrogen–deuterium exchange mass spectrometry (HXMS) experiments showed that, following DNA binding, the HD undergoes local unfolding, specifically in two of the helices (αB and part of αF), which otherwise prevent the active site from readily binding NAD+ [24]. Zn2 and BRCT were shown to be disposable for catalytic activation, somewhat ironically, since BRCT is a major site of automodification [25]. Nevertheless, in the absence of BRCT, the enzyme is still able to catalyze auto-PARylation. Solution NMR studies have shown that, while not necessary for activation, Zn2 plays an important role in DNA binding [26]. While DNA binding is critical for robust PARP-1 activation, PARP-1 binds to DNA in a sequence-independent manner [22]. PARP-1 binds to a wide variety of DNA structures, including blunt ends, 3′ and 5′ overhangs, with or without 3′ or 5′ modifications, and gapped DNA [20], as well as non-B form DNA such as DNA hairpins, cruciform DNA, and stably unpaired regions of DNA [27]. PARP-1 does show a preference for damaged DNA, preferentially binding a nicked DNA over circular, undamaged DNA [28]. While the kinetics and overall catalytic activity are affected by the DNA structure, PARP-1 activation can be triggered simply by the presence of exposed nucleotides and/or distortions in the helical backbone [27]. This provides a rationale for the involvement of PARP-1 in a wide variety of DNA repair processes [29].

During catalysis, PARP-1 chemically cleaves the nicotinamide moiety from nicotinamide adenine dinucleotide (NAD+). To avoid confusion, we will designate the two ribose groups as “adenosine-proximal” and “adenosine-distal” based on their location relative to the adenosine group, which remains present through the reaction (Figure 2, left). Within the PARP-1 active site, three chemically distinct reactions occur, all of which covalently link the adenosine-distal ribose of the NAD+ in the active site: (1) initiation, linkage to an amino acid receptor; (2) elongation, linkage to the adenosine-proximal ribose of the nascent ADP-ribose chain; (3) branching, linkage to the adenosine-distal ribose of the nascent ADP-ribose chain [30]. The rate-limiting step in this process is initiation, while elongation is >200-fold faster and processive [31,32].

Initiation primarily involves the HYE catalytic triad, consisting of H862, Y896, and E988 [33,34,35]. H862 is involved in substrate binding and positioning, forming a hydrogen bond with the adenosine ribose (Figure 2, right), and mutants show significantly lower affinity for NAD+ (H862A) [34] or a lack of detectable activity (H862D) [36]. Y896 plays a role in substrate binding by stacking with the nicotinamide moiety (Figure 2, right) [35], though no published articles have examined mutations to this residue. E988 is implicated in substrate positioning, forming a hydrogen bond with the nicotinamide ribose (Figure 2, right), and mutations E988Q and E988K inactivate the elongation activity of PARP-1 [30,33,34]. During initiation, the acceptor amino acid must be nucleophilic. Indeed, D and E residues have long been considered the primary acceptors of ADP-ribosylation, owing to both their chemical nature and mass spectrometry-based detection methods [37,38]. However, other amino acids, such as K, R, S, T, C, and Y, have been shown to be modified [15,38,39,40,41,42], suggesting PARP-1 has an underlying mechanism for the enzymatic deprotonation of amino acids, which are non-nucleophilic under physiological conditions. Many ADP-ribosyl transferases capable of modifying arginine residues, which are also not nucleophilic at physiological pH, contain a catalytic dyad of E-X-E [43], while PARP-1 only has a single E residue in this pocket. An accessory protein, histone PARylation factor 1 (HPF1), was shown to switch the substrate specificity from D/E to S residues [44] by inserting a catalytic glutamic acid residue, HPF1 E284, into the PARP-1 active site, allowing this to work with PARP-1 E988 to allow the efficient catalysis of Serine ADP-ribosylation [45,46]. Still, in mass spectrometry studies using highly purified recombinant PARP-1, without adding HPF1, serine, lysine, and arginine, ADP-ribosylation was observed [38,41]. As these residues are not nucleophilic at physiological pH, the question of how these studies were able to observe the modification of non-nucleophilic residues becomes pertinent. E988 is in the appropriate position to act as a catalytic base, yet PARP-1 E988 mutants retain a minor amount of catalytic activity [30,33,34]. At the time of those studies, the ability of PARP-1 to modify serine residues was not as prominent of an idea, and the ability of E988 mutants to retain some level of catalysis was thought to reflect the nucleophilic nature of the D/E residues. PARP-1 E988 mutants, E988Q, E988K, E988A, and E988D, maintained the ability to perform initiation, albeit less robustly [30,34]. Kinetic assays interrogating E988A, E988Q, and E988D showed some impact, albeit mild, of the E988 mutations on the rate of initiation, with WT > E988Q > E988D > E988A [34]. It is interesting to note that the length of the amino acid correlates well with the initiation rate, rather than the charge of the amino acid. Based on the differences in initiation rates for the different mutations, it is possible E988 plays a non-essential role in initiation. The mechanism by which PARP-1 catalyzes ADP-ribosylation on non-D/E residues, in the absence of partner proteins, remains elusive.

Elongation involves H862, Y896, and E988 again during the catalytic step. M890 and Y986 are also involved during elongation, helping stabilize the adenosine and the pyrophosphate moieties, respectively, on the previous ADP-ribose unit [35,47,48]. In this step, E988 is crucial. E988D forms PAR less robustly and at a slower rate. E988A showed the minor elongation of MAR into oligo-ADP-ribose (OAR) at the longest time point assessed. E988Q did not show elongation activity [34], nor did E988K in a separate study [30]. Thus, E988 is playing a more active role in this enzymatic process. Mutations to Y986 impacted elongation and branching, with average PAR chains sizes of ~11 for Y986S, ~15 for Y986H, and ~38 for wild-type [30]. Interestingly, Y986S had the same amount of branching detected, while Y986H showed a 15-fold increase in branching [30]. The mechanism for the involvement of Y986 in branching is unknown. The elongation and branching involve the same glycoside bond, but occur on different riboses on the nascent chain. This minor difference could be related to the acceptor ADP-ribose binding in the wrong (opposite) orientation. 

Whether linear or branched, the resulting enzymatic product is a PAR chain, which can impact enzymes in a covalent or non-covalent way. Non-covalently, PAR is a signaling molecule that recruits enzymes to sites of DNA damage, impacting their subcellular localization and allowing the efficient repair of single- and double-stranded breaks [29,49,50,51]. Essentially, enzymes involved in genome maintenance can have an intrinsic affinity for PAR, which can come through a PAR-binding motif (e.g., histones, p53, XRCC1, Lig3, DNA polymerase ε) [52], a macro domain (e.g., macroH2A, PARP-9, PARP-13) [53] containing a WWE domain (PARP-12, E3 ubiquitin-protein ligase (RNF146)) [54], or via a PAR-binding zinc finger domain (aprataxin polynucleotide kinase like factor (APLF), checkpoint with forkhead and ring finger domains (CHFR)) [55,56]. Alternatively, PAR can be covalently linked to target enzymes, which affects enzyme electrostatics, as an average PAR chain is ~38 ADP-ribose units in length [30], and there are two negative charges per ADP-ribose. Proteomics experiments have identified many protein targets for intracellular PARylation, primarily involved in chromosome organization, transcription, DNA repair, and mRNA processing [37]. Thus, PARylation can impact protein behavior by manipulating electrostatics and/or subcellular localization.

PARylation can be reversed by poly(ADP-ribose) glycohydrolase (PARG) [57] or another ADP-ribosyl hydrolase (such as ARH3, etc.) [53,56]. These enzymes can partially or fully remove the PAR chains from the modified molecule, and the balancing of PAR polymerase and PAR (glyco)hydrolase activities is required for many cellular processes, namely, DNA repair.

## 3. The Debate Regarding Mitochondrial PARP-1

PARP-1 is a prominent DNA repair enzyme in the nucleus, where it helps improve the efficiency of single-strand break repair and plays a role in other DNA damage response mechanisms [50]. The highly oxidative environment of the mitochondria leads to high levels of DNA damage, which must be quickly repaired, but how the organelle is able to efficiently clear DNA damage is unclear [5,6]. Some groups have explored whether PARP-1 plays a role in DNA damage repair in mitochondria, which we will discuss below. PARP-1 lacks a canonical mitochondrial localization sequence, so it had been unclear how PARP-1 could be imported into mitochondria. However, ADP-ribosylation has been observed in mitochondria, and much controversy has arisen regarding whether PARP-1 is the enzyme responsible for this observation. To parse through this, we will first describe the evidence supporting mitochondrial PARP-1, then the evidence against it.

To our knowledge, mitochondrial ADP-ribosylation was first described in rat liver mitochondria and the enzyme system reportedly generated protein-linked oligomeric ADP-ribose, though the enzyme class was not elucidated (e.g., NAD+ glycohydrolase vs. ADP-ribosyl transferase) [58]. Further studies would corroborate the finding of ADP-ribose in mitochondria, localizing it to the inner mitochondrial membrane, but the enzyme class was still unclear [59]. A mitochondrial ADP-ribosyl transferase was determined to be responsible for the protein ADP-ribosylation rather than the glycohydrolase, as reactions performed following the removal of the glycohydrolase still produced oligo-ADP-ribosylated proteins [60]. Similarly, ADP-ribosyl transferase activity was found in synaptic and nonsynaptic mitochondria purified from rat brain [61]. Building upon this work, a ~110 kDa protein capable of auto-ADP-ribosylation was described, but the authors were dissuaded from publishing these data, allegedly due to the general disbelief in mitochondrial ADP-ribosylation [62]. Later, the immunostaining of Sertoli and HeLa cells would show PARP-1 present in mitochondria at significantly higher concentrations than the surrounding cytoplasm [63]. The lack of any canonical mitochondrial localization sequence could explain the skepticism regarding mitochondrial PARP-1. Indeed, the question of how PARP-1 would enter the mitochondria was outstanding for many years. In 2009, Mitofilin, a transmembrane protein in the inner mitochondrial membrane, was identified to be responsible for transporting PARP-1 into mitochondria using immunoprecipitation–mass spectrometry, confocal laser microscopy and Western blotting methods; the depletion of Mitofilin abrogated PARP-1 mitochondrial import [64]. Other groups were able to find PARP-1 in mitochondria using proximity ligation assays, showing that PARP-1 interacted with both Exonuclease G (ExoG) and Pol γ in mitochondria [65]. Disease states, such as Chagas disease, or genotoxic stresses, like hydrogen peroxide treatment, can affect the concentration of PARP-1 in mitochondria [66]. The synthesis of a mitochondria-targeting PARP-1 inhibitor, XJB-Veliparib, inhibiting PAR formation in the mitochondria clearly illustrates the presence of PAR in mitochondria [67]. Liquid chromatography–mass spectrometry showed XJB-Veliparib specifically in mitochondrial fractions. Therefore, XJB-Veliparib, a specific PARP-1/2 inhibitor, was transported into the mitochondria, where it impaired intramitochondrial PARylation. The role of PARP-1 in mitochondria is being explored, and evidence supports a role for PARP-1 in mtDNA maintenance [64,65,66] and transcription [68].

On the other side of this argument, in a study investigating PARP-1-induced cytotoxicity in mouse embryonic fibroblasts treated with the DNA damaging agent N-methyl-N′-nitro-N-nitrosoguanidine (MNNG), the authors performed subcellular fractionation and Western blots, and only found PARP-1 in the nuclear fraction, but not the mitochondrial in both control and MNNG-treated cells [69]. Despite not finding PARP-1 in the mitochondrial fraction, the study clearly illustrated that PARP-1 activation led to apoptosis-inducing factor (AIF) being released from mitochondria, and therefore the nuclear PARP-1 was said to be impacting the mitochondria. In another study using HeLa cells to study the cellular impacts of PARP-1 hyperactivation following exposure to MNNG, the authors utilized similar experiments, also failing to find PARP-1 in mitochondria. This study added to the field that PARP-1 and PAR were exclusively nuclear, seeing no signal for either in the cytosol or mitochondria [70]. Although the authors only found PARP-1 in the nucleus, they noted a decrease in mitochondrial ATP concentration following PARP-1 activation, again suggesting that the activation of nuclear PARP-1 impacts the mitochondria indirectly. A study exploring the localization of PARG in mouse cortical neurons, using the same major experiments for determining subcellular localization, only saw PARP-1 in the nucleus as well [71]. Similar experiments were performed on HeLaS3 and HEK293 cells [72], and also saw no PARP-1 in mitochondria or cytosol, but did observe oligo-ADP-ribose in mitochondria. More recently, it was shown that mitochondrial PARylation did not decrease in U2OS cells treated with inhibitors specific for PARP-1/2, and that U2OS PARP-1 knockout cells showed an increase in mitochondrial PARylation relative to the control [73]. In this study, the presence of PAR in mitochondria is established, contradicting the above studies. It was not stated that the subcellular localization for the PARP inhibitors was evaluated, and whether those inhibitors would be taken into the inner mitochondrial matrix is unclear. While all of these studies show no PARP-1 in mitochondria, they imply an intricate crosstalk between the nucleus and the mitochondria, whereby nuclear PARP-1 activation has an impact on mitochondrial homeostasis. Some of these studies utilized MNNG to damage the DNA, but whether this is transported into the inner mitochondrial matrix and damages the mtDNA at a rate comparable to nuclear DNA is not discussed. Additionally, these studies do not agree on the degree of mitochondrial ADP-ribosylation. Studies using DNA-damaging agents that specifically damage mitochondria DNA would be highly informative for the argument against PARP-1 in mitochondria.

Currently, the field is divided regarding whether PARP-1 localizes to mitochondria. Many studies utilize imaging-based techniques, such as the generation of fusion proteins followed by fluorescence microscopy or immunofluorescence. Experiments in which cells are transfected with vectors expressing exogenous fusion proteins are blind to the subcellular localization of the endogenous protein, and can result in artifacts caused by protein overexpression. In immunofluorescence experiments, artifacts can arise based on antibody quality, epitope availability in the cell (e.g., if a binding partner covers up the epitope), and the fixation method used. A systematic study comparing the subcellular distribution of proteins using both immunofluorescence and fusion proteins found that 82% of proteins tested showed “similar” localizations, with “similar” being defined as “one localization observed with both methods but with additional localizations in either of the two methods” [74]. For this reason, some authors decided to avoid imaging-based techniques and analyze PARP-1 subcellular localization using highly purified mitochondria [66,68]. Even in these cases, when Western blots are used to determine subcellular localization, artifacts can occur due to antibody quality and titer.

The debate regarding whether PARP-1 localizes to the mitochondria continues, as evidenced by the number of the above articles that were written within three years of this one. Some studies positively identify PARP-1 in mitochondria, while others do not see this enzyme localized to the mitochondria organelle under their experimental conditions. The identification of a transmembrane protein that imports PARP-1 into the mitochondria, Mitofilin, and the data suggesting that PARP-1 impacts the repair and transcription of mtDNA provide convincing evidence for PARP-1 in mitochondria. These data are bolstered by the in situ assays showing an interaction between PARP-1 and mitochondria-specific proteins within the mitochondrial matrix. The mechanisms governing the PARP-1/Mitofilin interaction remain unexplored, and would provide insights into the conditions required for PARP-1 import into mitochondria. The studies in which PARP-1 was not seen in mitochondrial fractions could suggest that PARP-1 import requires specific cell types and/or specific conditions, as opposed to being constitutively found in mitochondria.

## 4. Base Excision Repair

Base excision repair (BER) is the primary mechanism for repairing oxidatively damaged DNA bases. As the simplest mechanism of DNA repair, BER is also a useful model for understanding genome maintenance. There are five fundamental steps: (1) removal of the damaged base, (2) incision of the DNA backbone, (3) removal of the lesioned DNA backbone (resultant of step 2), (4) filling of the gapped DNA, and (5) ligation of the nicked DNA. The product of each step provides the substrate for the following step, until ligation is completed [75].

Figure 3 depicts the distinct enzymatic steps (listed above) required for BER to be completed, with examples of enzymes reportedly responsible for each of these steps for both the nuclear and mitochondrial genomes. Step 1 is carried out by a DNA glycosylase, which can be monofunctional or bifunctional. Monofunctional DNA glycosylases (ex: uracil DNA glycosylase; UNG) strictly remove the damaged base, leaving an apurinic/apyrimidinic (AP) site [76]. Step 2 is then carried out by an AP endonuclease, such as AP endonuclease 1 (APE1), resulting in a 3′-OH and a 5′deoxyribose phosphate (dRP) [7]. Bifunctional DNA glycosylases can remove the damaged base and excise the DNA backbone, accomplishing both steps 1 and 2. Examples of bifunctional DNA glycosylases include 8-Oxoguanine DNA Glycosylase 1 (OGG1) and Nei Like DNA Glycosylase 1 (NEIL1), which perform β and β,δ-elimination reactions, respectively [77,78,79].

Various enzymes can perform step 3, depending on which glycosylase was involved in processing the damaged DNA base. A 3′P, which can occur following NEIL1 activity, can be cleared by polynucleotide kinase (PNK), yielding a 3′-OH. The β-elimination reaction pathway performed by OGG1 yields a 3′deoxyribose phosphate (dRP), and APE1 can remove this moiety to yield the 3′-OH required for gap-filling. For both scenarios, the resulting DNA damage enters the single nucleotide (SN) BER subpathway, where a DNA polymerase, such as DNA polymerase beta (Pol β), can replace the missing nucleotide and ligation can occur.

Biochemically, the removal of a 5′dRP—which can be produced following UNG and APE1 activity—is the rate-limiting step in this BER subpathway [80,81]. The processing of the 5′dRP occurs differently in the nucleus than it does in mitochondria. In the nucleus, the oxidation state of the 5′dRP determines the subpathway for the gap-filling step. In the unoxidized form, DNA polymerase beta (Pol β) performs dRP lyase activity, removing the lesion [82,83]. Pol β itself can then replace the single-nucleotide (SN) gap, and ligation can occur, concluding the SN-BER subpathway. Oxidation of the 5′dRP generates a 5′deoxyribonolactone (dL), which Pol β cannot process as it forms a covalent protein–DNA crosslink [84,85]. Thus, long-patch (LP)-BER is carried out, where DNA polymerase delta (Pol delta), aided by proliferating cell nuclear antigen (PCNA), performs strand-displacement synthesis, generating a 5′DNA flap that contains the dL lesion, which is removed by flap endonuclease 1 (FEN1) [86]. Following flap removal by FEN1, ligation can occur. In both SN- and LP-BER, the DNA Ligase 3 (Lig3)/X-ray repair cross-complementing protein 1 (XRCC1) complex is primarily responsible for this step in the nucleus [87].

In mitochondria, the oxidation state of the dRP appears to play a lesser role in determining the subpathway, as ExoG can efficiently process both 5′dRP and 5′dL lesions [88]. The exonucleolytic register of ExoG is 2-nt, resulting in a multi-nucleotide gap [89,90]. By definition, LP-BER incorporates ≥2 nucleotides, for which Pol γ is an effective gap-filling polymerase [91]. The ligation of the nicked DNA is still performed by Lig3, yet Lig3 acts independently of XRCC1 in mitochondria, despite requiring XRCC1 in nuclei, because XRCC1 is not localized to mitochondria [92]. The BRCT domain of Lig3 and one of the BRCT domains in XRCC1 mediate the formation of a relatively compact, stable heterodimer [93,94]. XRCC1 is a scaffolding protein, which interacts with Pol β [95], PARP-1 [96], APE1 [97], and polynucleotide kinase (PNK) [98] through distinct regions of the protein, helping to associate the repair factors at sites of DNA damage [99]. Considering this, the report that Lig3 acts independently of XRCC1 in mitochondria is quite intriguing as the protein partners for Lig3 would likely be different in the absence of XRCC1.

DNA ligation occurs via a three-step catalytic mechanism: enzyme adenylation, transfer of the adenyl group onto DNA, and phosphodiester bond formation [100]. Abortive ligation products arise when only the first two steps of this mechanism occur, resulting in a 5′AMP on the gapped DNA [101]. In both nuclei and mitochondria, aprataxin (APTX) plays a key role in removing the abortive ligation product and allowing the proper joining of the DNA ends [102,103].

Another enzyme involved in the maintenance of nuDNA and mtDNA is tyrosyl-DNA-phosphodiesterase 1 (Tdp1), which helps clear the tyrosyl–DNA lesion left from a covalently trapped topoisomerase–DNA complex that has been proteolytically digested down to the terminal tyrosine [104]. By clearing these covalent protein–DNA crosslinks, Tdp1 promotes mitochondrial gene expression [105].

The biggest debate regarding mtBER enzymes regards the involvement of Pol β. Pol β removes dRP groups and is better at filling 1-nt gaps than Pol γ [106]. Evidence has been presented for Pol β localization in mitochondria [107,108], while others have argued that the Pol β activity originally attributed to the mitochondrial fraction was due to contamination with polysomes, microsomes, and liposomes. Upon the further purification of mouse liver mitochondrial extracts, the authors determined that the mitochondria was free of Pol β-like activity [109]. Work by the Bohr lab has added another layer to this debate, with evidence showing that the presence of Pol β in mitochondria is cell-type specific. In agreement with [109], Pol β was not detected in mitochondria from mouse liver, heart or muscle, but was found in that of mouse brain and kidney. Using their model HEK293T cells, Pol β was found to interact with DNA maintenance proteins specific to mitochondria, namely, Twinkle, mitochondrial transcription factor A (TFAM), and single-stranded DNA binding protein (SSBP). The authors also found Pol β interacting with chaperones which aid in protein folding following mitochondrial importation, namely, heat shock protein family A (Hsp70) member 9 (HSPA9), heat shock protein family D (Hsp60) member 1 (HSPD1) and GrpE like 1 (GRPEL1) [110]. This study provides an excellent basis for context-specific Pol β importation into mitochondria and supports the idea of Pol β impacting mtDNA maintenance. The aspects of mtDNA maintenance in which Pol β participates remain an open question. Specifically, Twinkle and TFAM are important in the replication and transcription of mtDNA. Pol β is a known DNA repair polymerase, so it may play a role in mtBER, but which subpathways involve Pol β, and the step at which Pol β is involved, needs to be elucidated. For the purposes of this review, we consider Pol γ as the major enzyme responsible for the gap-filling activity (step 4) in mitochondrial LP-BER.

## 5. The Role of PARP-1 during DNA Repair

While DNA strand breaks are common, the relative percent of damaged DNA versus undamaged DNA is quite low. PARP-1 is heralded as a DNA damage sensor, rapidly co-localizing at sites of DNA damage in live cell models following laser-induced damage [111,112]. Structurally, PARP-1 contacts the exposed nucleotides and the minor groove of the DNA helix [22,26]. These two features are common among all types of DNA damage, which helps explain the promiscuity of PARP-1 in binding to damaged DNA. Once bound to DNA, the allosteric signaling mechanism ultimately allows NAD+ access to the active site and catalysis can occur, generating the PARylation-dependent recruitment of downstream repair enzymes, such as XRCC1, which acts as a loading platform [111]. This mechanism implies that PARP-1, prior to DNA damage, exists in isolation, separate from other repair factors. However, PARP-1 has also been found to exist in pre-formed complexes with such enzymes. Using mouse embryonic fibroblasts (MEFs) and the “affinity capture” of Pol β, the Wilson group characterized a multi-protein complex consisting of Pol β, PARP-1, Lig3, XRCC1, Tyrosyl-DNA Phosphodiesterase 1 (TDP-1), Polynucleotide Kinase (PNK), and Replication Protein A (RPA). The authors noted the absence of glycosylases as well as Apurinic/apyrimidinic Endonuclease 1 (APE1) and Flap Endonuclease 1 (FEN1) [113]; therefore, instead of being a fully functional “repairosome”, this constitutes a multi-protein complex capable of performing many BER steps. For brevity, this complex will be referred to as the PACC (Pol β Affinity Capture Complex). Owing to the presence of XRCC1, which is accepted to not localize to mitochondria [92], we believe the PACC is a largely nuclear complex. A mitochondrial complex of similar capabilities could be beneficial for mtBER. The mitochondrial DNA is packed into a nucleoid structure, as opposed to a chromosome [114,115], making it unclear whether a pre-formed complex would be necessary for maintaining the mitochondrial genome. Intriguingly, an immunocomplex isolated from mitochondria was shown to carry out LP-BER in vitro, which would support the existence of a pre-formed complex. The complex contained APE1, Lig3, and both subunits of Pol γ. Notably, FEN1 was not detected, and the authors concluded that other proteins must be involved in this complex, including a 5′ endo/exonuclease [116].

On the basis that PARP-1 was part of the PACC [113] and was shown to interact with Pol γ and ExoG in A549 cells [65], PARP-1 could be involved in the mitochondrial repairosome. The above study did not evaluate whether PARP-1 was present in their immunocomplex. To evaluate the possibility of PARP-1 being involved in the mitochondrial repairosome, we will review the repair enzymes that PARP-1 has been reported to interact with, in the same order as the repair pathway is obligated to occur.

Step 1: Removal of the damaged base. Oxidative damage to guanine results in 8-Oxoguanine, which must be removed by a glycosylase, such as 8-Oxoguanine DNA Glycosylase 1 (OGG1). To our knowledge, it has not been shown whether PARP-1 would bind a DNA substrate in which the only damage is a single base. However, PARP-1 was shown to directly bind OGG1 in mouse liver and brain, as well as HeLa cells [117]. DNase treatment did not abrogate the interaction, and deletion-based experiments using recombinant proteins allowed localization of the interaction to the BRCT domain of PARP-1 and amino acids 79–180 of OGG1. Interestingly, some interaction between OGG1 and the DNA binding domains of PARP-1 was observed, but only in a DNA-dependent manner, implying both DNA-mediated and DNA-independent interactions can occur. The full-length proteins still interacted following DNase treatment. The PARP-1/OGG1 interaction had functional consequences: the presence of OGG1, but not a bacterial glycosylase with the same biochemical activity, stimulated histone PARylation when a DNA substrate capable of stimulating PARP-1 activity was present. The inhibition of OGG1 activity was noted specifically with activated PARP-1 (e.g., in the presence of NAD+), rather than just the physical presence of PARP-1 or PAR. When tested in MEFs, OGG1 −/− MEFs subjected to hydrogen peroxide treatment showed decreased PAR synthesis, while WT MEFs showed an increase in PAR, indicating the OGG1-dependent stimulation of PAR synthesis. Further experiments revealed that HeLa cells treated with PARP-1 inhibitor showed no significant difference in levels of 8-oxo-G. Adding a DNA damaging agent showed an increase in 8-oxo-G levels, yet this reverted to the same level as the control when both the DNA damaging agent and PARP-1 inhibitor were included. This study provides solid evidence for a PARP-1/OGG1 interaction in vitro and in vivo. The mechanism behind this interaction and the regulators would require further study. Intriguingly, OGG1 was noted to be absent from the PACC, which contained PARP-1. Thus, the two articles are not in perfect agreement. Differences in experimental conditions could account for the lack of agreement, or a context-specific PARP-1/OGG1 interaction. Additionally, in the context of the PACC, the PARP-1 BRCT domain, which contributes directly to the PARP-1/OGG1 complex, may not be accessible for the binding of OGG1. PARP-1 is an abundant enzyme, with concentration estimates of up to 20 µM in the nucleus [118], making it conceivable that both situations could be true, with a copy of PARP-1 bound to OGG1, while a separate copy is bound to the PACC. Similar observations were made regarding a PARP-1/NEIL1 interaction [119], though interactions between PARP-1 and monofunctional DNA glycosylases were not found in our search of the literature.

Step 2: Incision of the DNA backbone. The removal of the damaged base results in an apurinic/apyrimidinic (AP) site, and an AP endonuclease, such as APE1, then incises the phosphodiester backbone of DNA adjacent to the AP site. Cell-based evidence for an OGG1/APE1 or PARP-1/APE1 interaction is lacking, and both glycosylases and endonucleases (including OGG1 and APE1) were absent from the PACC. However, detailed kinetic experiments using recombinant proteins have shown a functional relationship between OGG1 and APE1 in which APE1 increased the K_cat_ of OGG1 without forming stable binary OGG1/APE1 or ternary OGG1/APE1/DNA complexes [120]. This implies a synergistic effect without complex formation. Similarly, a functional interaction between PARP-1 and APE1 has been explored. These studies have shown PARP-1 can bind to AP sites [121], representing the first DNA repair intermediate in which there is clear biochemical evidence for PARP-1 binding. As such, detection of the AP site could be the first step in BER in which PARP-1 becomes involved. A study comparing the binding affinity of PARP-1 for the different DNA repair intermediates could shed light on the steps that PARP-1 is likely involved in. Functionally, the PACC showed weak BER activity in the absence of APE1, and the presence of APE1 greatly increased this activity. Experiments with isolated enzymes determined that PARP-1 stimulates APE1 strand incision in a PARylation-independent manner. PARP-1 and APE1 co-localize at AP sites using DNA tightrope assays [121], though the frequency of this occurrence was unclear and context-specific. Importantly, these single molecule experiments suggest that the co-localization of APE1 with PARP-1 at AP sites facilitates PARP-1 diffusion along the DNA, prior to finally dissociating once it is highly modified. The findings from these studies show PARP-1 has different modes of interacting with repair enzymes: PARP-1 inhibited OGG1 when activated, yet stimulates APE1 without being activated.

Step 3: Removal of the lesioned DNA backbone. Following incision, a nicked DNA substrate containing a 5′dRP and a 3′OH is generated. In the nucleus, Pol β is the major dRP lyase [82] and has been shown to interact with PARP-1 in MEFs and in vitro. Biochemical experiments using recombinant PARP-1 and Pol β suggested that PARP-1 does not impact the dRP removal activity of Pol β [122]. Thus, PARP-1 can interact with the enzyme responsible for the majority of dRP removal in the nucleus without affecting this specific enzymatic activity.

In mitochondria, the enzyme responsible for dRP removal is debated, and recent work from our lab suggests that both dRP and dL sites can be excised by ExoG [88]. Using proximity ligation assays and immunoprecipitation, ExoG was shown to interact with, and be PARylated by, PARP-1 in A549 cells, and this interaction was enhanced by oxidative damage [65]. No published studies have shown whether PARP-1 impacts the function of ExoG, so currently there is no evidence suggesting PARP-1 would impact the enzymatic dRP removal. An examination of the dRP removal kinetics of Pol β and biochemical assessment of the PARP-1/ExoG interaction would add valuable information towards elucidating whether PARP-1 has an impact on dRP removal.

Step 4: Filling of the gapped DNA. The removal of the dRP from the DNA backbone results in, minimally, a 1 nt gap. These single nucleotide gaps are generated in the nucleus upon Pol β-mediated dRP removal. This can then be filled by a repair DNA polymerase, such as Pol β itself, which adds a single nucleotide to the DNA backbone, successfully completing this step. In vitro, PARP-1 seems to permit single-nucleotide incorporation by Pol β [123], though the experimental conditions involved pre-forming Pol β/DNA complexes on ice prior to the addition of PARP-1. It is not explicitly stated whether PARP-1 was added sequentially or simultaneously with the dNTP mixture required for gap-filling, making it unclear whether PARP-1 was complexed with Pol β prior to nucleotide binding. This is an important detail, since PARP-1 and Pol β would be complexed prior to encountering DNA in the context of the PACC. Pol β also has the ability to perform strand displacement synthesis, which then generates a 5′ DNA flap that requires removal. Pol β strand displacing synthesis is stimulated specifically when PARP-1 and FEN1 are both present, but in the absence of FEN1, PARP-1 strongly inhibits the strand displacement synthesis of Pol β [124]. As FEN1 is absent from the PACC, we speculate that Pol β would not be performing strand displacement synthesis in the context of the PACC.

In mitochondria, ExoG’s ability to remove the dRP would result in a multi-nucleotide gap, and LP-BER must occur. Pol γ has the ability to efficiently fill the gapped DNA [91], and cell-based and mouse-based experiments have examined a PARP-1/Pol γ interaction [65,66]. Szcesny et al. [65] saw an increase in genome integrity when PARP-1 was depleted, and Wen et al. [66] found that Chagasic mice displayed the overexpression of PARP-1, which negatively impacted mitochondrial genome integrity in a Pol γ-dependent manner. The studies agreed that PARP-1 could inhibit mitochondrial BER under certain conditions. When the PARP-1/Pol γ interaction was reconstituted in vitro, PARP-1 was found to PARylate and regulate Pol γ in an NAD+-dependent manner. This regulation occurred via a ternary complex between Pol γ, PARP-1, and a gapped-DNA. Pol γ gap-filling activity was strongly inhibited until sufficient PARylation had occurred, and the same results were seen when experiments were performed using Pol β. These experiments used a DNA substrate containing a 3 nt gap, which specifically mimics the LP-BER pathway. These experiments allowed PARP-1 and Pol γ (or Pol β) equal opportunity to bind DNA, rather than adding PARP-1 to a preformed polymerase:DNA complex. Under these conditions, DNA extension was not observed in the absence of NAD+, indicating the physical presence of PARP-1 prevents DNA extension, and PARylation regulates gap-filling DNA synthesis. PARP-1 did not exert a regulatory effect on Pol γ when a primer/template DNA substrate, mimicking a DNA replication start site, was used [125]. This is likely due to a lack of proper orientation on a primer/template DNA, which lacks a downstream dsDNA portion. The NAD+-dependent regulation of gap-filling suggests PARP-1 and PARylation may serve as a checkpoint for DNA repair, prohibiting BER when insufficient NAD+ levels are present. The fact that PARP-1 and PARylation regulated Pol γ and Pol β could indicate this checkpoint is not unique to mitochondria. As DNA replication was not regulated by PARP-1 or PARylation, this could imply that mitochondrial genome replication is favored over mitochondrial genome repair when NAD+ levels are depleted. This concept can be further investigated using mitochondria-specific PARP-1 inhibitors, which are currently being developed [67].

Step 5: Ligation of the nicked DNA. Once the gap-filling has been performed, regardless of which sub-pathway is utilized, a nicked DNA is generated. The ultimate step of DNA repair is ligating this nicked DNA. In the nucleus, the Lig3/XRCC1 complex accomplishes this task. This complex interacts with PARP-1 in the context of the PACC. HeLa cells provided in situ evidence suggesting the overexpression of XRCC1 decreased PARP-1 activity [96]. The same study showed that PARP-1 and XRCC1 interact via their BRCT domains and the DNA-binding domains of PARP-1 in vivo. Recently, studies using RPE-1 cells showed that XRCC1 helps to reduce the cytotoxicity of PARP-1 during BER by constantly competing with PARP-1 for DNA intermediates [126]. In vitro evidence supports XRCC1 binding to PAR chains [127], which would help bring Lig3 in close proximity to the nicked DNA. The PARP-1/XRCC1 interaction has been shown in cell models and using recombinant proteins, and this interaction would impact the ligation step in a PARylation-dependent manner.

Lig3 was shown to play an important role in mitochondria, independent of XRCC1 [92]. PARP-1 was found to directly interact with Lig3 via immunoprecipitation–mass spectrometry, and two hybrid assays indicated this interaction occurred through the zinc finger domains of each protein. PARP-1 functionally regulates Lig3, displaying a strong inhibition of ligation in the absence of NAD+. In the presence of NAD+, Lig3 regained some ligation abilities (~40–70%, depending on isoform). PAR itself did not inhibit Lig3, indicating PARP-1′s impact on Lig3 is largely due to the physical presence of PARP-1 rather than PARylation activity. Interestingly, Lig1 regained complete ligation ability [128]. This may indicate Lig3, but not Lig1, is PARylated by PARP-1, given that following the PARP-1 PARylation of Pol γ, only ~60% of activity returned [125]. The effects of PARP-1 and PARylation on Lig3 were incredibly similar to those seen in our study of the PARP-1/Pol γ interaction, and the mechanism of interaction may be similar.

A graphical summary of the role of PARP-1 in LP-BER, specifically in mitochondria, is presented in Figure 4, based on the studies described above. This working model requires three copies of the PARP-1 molecule, all of which end up modified. However, if a pre-formed complex between PARP-1, Pol γ, and Lig3 does exist, the number of required PARP-1 copies could decrease. In lieu of experimental evidence for this complex, we show three active copies of PARP-1.

First, PARP-1/OGG1 binary complexes are formed via the PARP-1 BRCT and amino acids 79–180 of OGG1 [117]. We anticipate that the affinity for this lesion comes largely from OGG1, as it is unknown whether PARP-1 can bind this type of DNA damage. OGG1 produces an AP site, which can activate PARP-1 [113,121]. PARP-1 activation inhibited OGG1, and Western blots in the supplement suggest PARP-1 PARylates OGG1 [117]. The presence of PARP-1 at an AP site stimulates APE1 incision, and the co-localization of APE1 and PARP-1 at an AP site induces PARP-1 diffusion along DNA and subsequent release from DNA once PARP-1 is highly modified [121]. It was stated that Pol beta performs dRP removal, unimpeded by PARP-1 [122], thus we infer that PARP-1 does not impact this step of mtLP-BER, and ExoG dRP removal is unaffected by PARP-1. During gap-filling, the presence of PARP-1 is strongly inhibitory towards Pol γ (and Pol β), only releasing when a sufficient level of NAD+ is present. Once the NAD+ requirement is met, both auto- and trans-PARylation occurs, followed by gap-filling and the dissociation of both enzymes from DNA. Lastly, Lig3 must ligate the gapped DNA, and this activity is stunted by the presence of PARP-1 in the absence NAD+ [128]. This could serve as a second checkpoint for mtLP-BER. Once sufficient NAD+ is met, or if the PARP-1 concentration is low, ligation can occur.

## 6. Concluding Remarks

The ability of PARP-1 to complex with and modify target proteins makes it an interesting topic for biochemical research. PARP-1 can bind several types of damaged DNA, allowing it access to many different repair enzymes during this process. Recent studies showing that PARP-1 impacts BER in an NAD+-dependent manner have reinforced the importance of understanding the impact of PARP-1 on mitochondria, as the difference in mitochondrial and nuclear NAD+ levels could impact the behavior of PARP-1 in BER in each compartment. Currently, it is not known whether a bona fide “repairosome” exists in mitochondria. The impact of PARP-1 on several different steps in LP-BER makes it an attractive probe for examining this topic, especially with studies supporting a PARP-1/ExoG interaction [65], a PARP-1/Pol γ functional interaction [65,66,125], and a direct PARP-1/Lig 3 interaction without use of XRCC1 [128], as all of these are specific to mtBER.

## Figures and Tables

**Figure 1 biomolecules-13-01195-f001:**
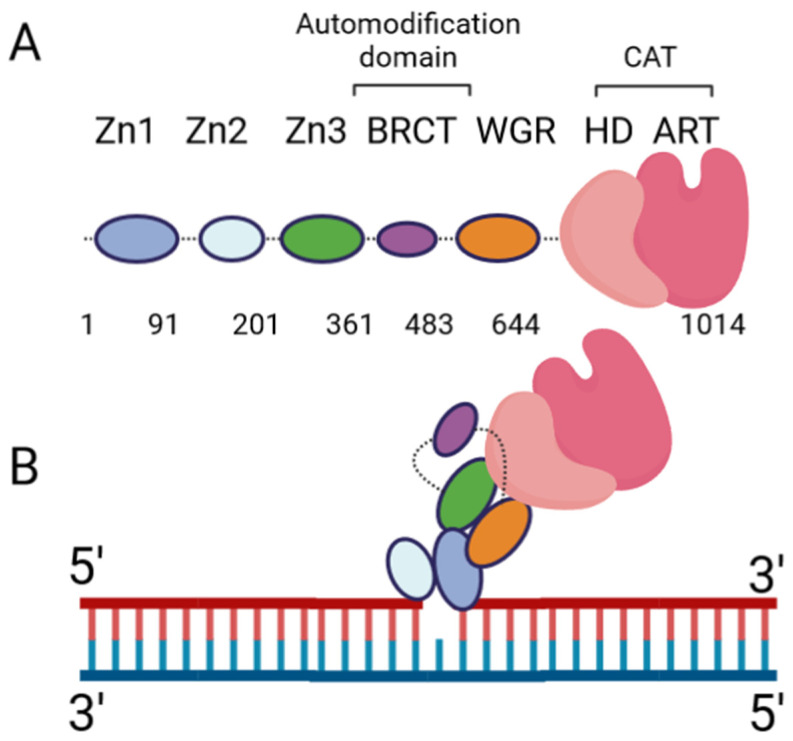
PARP-1 domain layout and arrangement in the DNA-bound form. (**A**) Cartoon representation of the six PARP-1 domains, connected by flexible linkers (dotted lines). (**B**) Cartoon representation of PARP-1:DNA complex, using a single-nucleotide-gapped DNA. Zn2 binds the 3′ end of the gap, while Zn1 binds the 5′ end. Zn1 serves as a scaffold for the assembly of the remaining domains in the appropriate orientation for catalytic activation.

**Figure 2 biomolecules-13-01195-f002:**
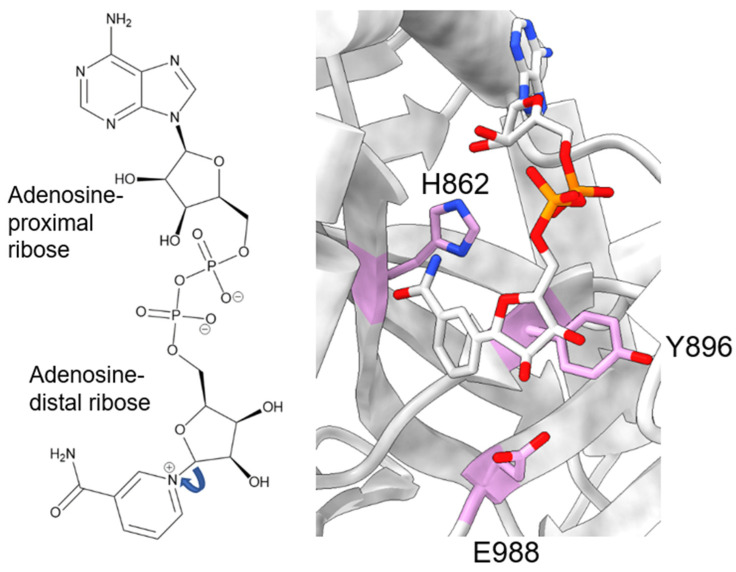
NAD+ molecule and its position in the PARP-1 active site. Left: Chemical structure of NAD+, with the two ribose moieties labeled as described in the text. Right: Zoomed-in view of a NAD+-analogue (benzamide adenine dinucleotide) bound to the PARP-1 active site (PDB: 6BHV). The H-Y-E triad of PARP-1 is shown in plum.

**Figure 3 biomolecules-13-01195-f003:**
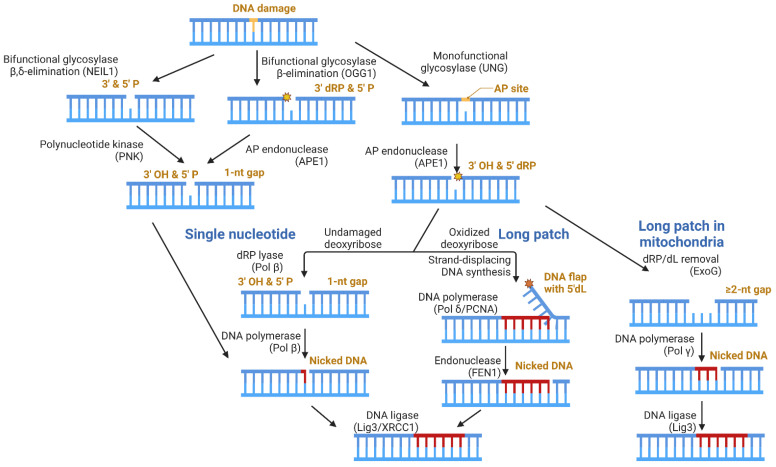
Cartoon depicting the five basic steps of BER. The enzymes responsible for steps 1 and 2 are shared between the two organelles, but the pathways diverge at step 3, during which the nucleus utilizes a traditional dRP lyase, while mitochondria use an exonuclease to remove the dRP. Steps 4 and 5 are chemically the same between the two compartments but carried out by different enzymes or enzyme complexes.

**Figure 4 biomolecules-13-01195-f004:**
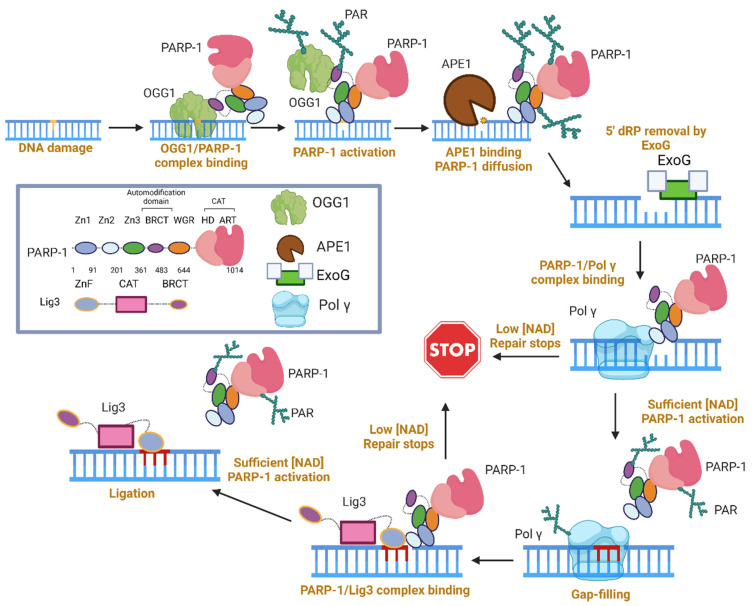
Working model for PARP-1 involvement during mtLP-BER.

## Data Availability

Not applicable.

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
