# Peer review of "The Role of Poly(ADP-ribose) Polymerase 1 in Nuclear and Mitochondrial Base Excision Repair"

_biomolecules, 2023, doi:10.3390/biom13081195_

Round 1

Reviewer 1 Report

In the review, Herrmann and Yin summarize the difference between mitochondrial (mt) and nuclear (nu) Base Excision Repair (BER) pathways, with a particular focus on the involvement of PARP-1 in mt- and nuBER and its functional interplay with other BER enzymes.

Although BER in the nucleus and the mitochondria is an exciting topic, the authors seem biased and uncritical regarding the current literature and thus miss providing a well-balanced review (particularly regarding the localization of PARP1 in the mitochondria). Enclosed are a few suggestions for improvement, although the list below is incomplete.

General criticism:

1) Chapters 1-3 need substantial revisions regarding style, spelling, and language in general (by a native-speaking person). Chapters 1 and 2 combine information already summarized elsewhere (lack of novelty) and thus can be omitted. Chapter 4 and the following chapters are better written (different author?) and provide a much better objective description and argumentation, including an in-depth explanation of all combined evidence.

2) While a non-essential role of PARP1 in BER has been reported (https://www.nature.com/articles/nrm.2017.53), its localization in the mitochondria is highly controversial since many published reports could not be reproduced. Thus, it is not enough to list these reports without critically discussing the experimental details and possibilities of why they differ in their conclusions, particularly in the view that the authors continue propagating a role of PARP1 for mtBER. Thus, the review would greatly profit from revisions regarding how the findings are presented.

3) Some of the used references are outdated and should be replaced (e.g., reference 12 from 2004!).

4) The authors should not only list all available literature on mitochondrial PARP1 but critically assess the experimental quality of the cited studies (e.g., IP from whole lysates (even from isolated mitochondria) do not strengthen the conclusion that a protein localizes to mitochondria, since it might bind to the outer membrane). Literature with low quality should not be further propagated. They also ignore multiple information sources to make a good argument for a controversial topic.

5) Surprisingly, the authors omit that PARP1 KO cells do perform BER both in the nucleus and mitochondria (step1-5 and Fig.4), suggesting that there is a functional compensation (e.g., by PARP2), or more likely that PARP1 is not essential for BER (https://www.nature.com/articles/nrm.2017.53).

Specific comments:

Line 8: PARP1 does not link PAR to proteins but catalyzes the synthesis of ADP-ribose polymers as a transferase.

Line 9: signaling molecule (not molecular).

Line 9: “More studies have supported” is a wrong statement and needs to be revised.

Line 18-25: The authors should include more references for the statement made (ideally from different laboratories).

Line 20: Including that the mitochondrial genome is a circular molecule is essential information that should be included.

Line 28: “This could argue” – language.

Line 38: provide valuable insights.

Line 42: for repairing

Line 49: delete “discussed below”.

Line 51: Page 1, The human ADP-ribosyltransferases (ARTs) superfamily has

The abbreviation ”PARPs” should no longer be used and should be replaced by ARTs (see https://doi.org/10.1111/febs.16142). This also refers to the abstract.

Line 55: “Two have no catalytic activity” This statement is wrong. PARP9 was shown to be ADP-ribosylated ubiquitin.

Line 61: “is” lethal.

Line 67: Surprisingly, the authors talk about PARP1 and PARP2 but only show PARP1 in Figure 1.

Line 77: The term “apo-enzyme” describes the protein part of an enzyme. The non-protein part cofactor, together with the protein part apoenzyme, forms a holoenzyme. It is not correct to define PARP1 as an apo-enzyme since it was shown to be also activated without DNA.

Line 84: write out 4 out of 6.

Line 86: Fig1B does not show the direct interface with CAT domain.

Line 106: Rewrite the sentence; “Upon PARP1 activation…” or “During PAR formation,…”

Line 133: Why did the author exclude explaining the catalytic triad?

Line 139: inactivate instead of inactive.

Line 143: There are so many different publications from different laboratories on the ADP-ribose acceptor site topic that should be acknowledged.

Line 145: mentioning non-nucleophilic amino acids as a reason and excluding cysteine is biased.

Line 149: It was also shown that PARP1 targets lysine without HPF1.

Line 179: involvement of.

Line 181: use “opposite”.

Line 185: The authors should include the repair of double-strand breaks as part of PARP signaling (perhaps even more critical than single-strand break repair) and not only single-strand breaks.

Line 198: While not wrong, most cited evidence is rather old. The authors might consider building upon historical evidence and newer evidence and techniques.

Line 210: The authors refer to an article that describes an auto-ADP-ribosylated 110 kDa protein isolated from mitochondria found in a thesis; however, this data is not available online and cannot be reviewed.

Line 214: The reported data were observed only with a self-made PARP1 serum, did not work in other cell lines, and did not have a mitochondrial mask to localize the signal.

Line 223: The IF pictures of the cited report are not convincing (slightly above the cell's background).

Line 254: The cited publications do not mention PAR but ADP-ribosylation and probably talk about mono-ADP-ribosylation.

Line 314/315: the listed XRCC1 interactors were not observed using mitochondrial lysates. Since this statement is made in the section about BER in mitochondria, it is very misleading and might lead to wrong conclusions for the reader.

Line 378 and the following: Many of the listed findings (steps 1-5) are unrelated to mitochondria and should not be listed after introducing the mitochondrial repairosome.

See general criticism point 1.

Author Response

Thank you for your careful review.  Please see our responses in the attached file.

Reviewer 2 Report

Herrmann and Yin provide an overview of PARP1 in nuclear and mitochondrial base excision repair. Overall, the paper is well written and brings up recent interesting observations on PARP1 in nuclear and mitochondrial repair. Whereas the review gives a very good impression, addressing the following shortcomings might improve it even further.

Line 47: change ‘present’ to ‘observed’ – the repair pathway could still be there even though scientists have failed to find the right conditions to detect it.

Section 2 provides an overview of the enzymatic mechanism of PARP1, and how it differs from PARP2 (line 68). PARP3 was however also mentioned a few lines up (line 59), which leaves the reader wondering if and how that enzyme differs from PARP1 and 2. It would be great if the authors could elaborate on PARP3 as well in this section, or outline the relevant knowledge gaps for this enzyme. Alternatively, they could explain why they don’t consider PARP3 relevant.

Line 103: ‘presence exposed nucleotides’ lack a preposition.

Section 3 outlines a debate for whether PARP1 is in mitochondria or not. This section would be strengthened immensely by a very short introduction explaining why this is important. Put bluntly, why should the reader care if PARP1 is in mitochondria or not? The authors proceed to scholarly list different studies arguing for and against PARP1 in mitochondria, and suggest that differences in experimental conditions might be responsible for the discrepancies in the literature. But I am not sure what the take-home message is for the reader. What is the authors’ best explanation for these discreptancies? Is mitochondrial PARP1 only detected in some cells and not others? Could different detection technologies explain the differences (e.g., antibody epitopes)? How should the field progress in the future to bring clarity ? Summarizing observations in a helpful table might bring more clarity to the reader.

Line 207: ‘The glycohydrolase’ is only mentioned at line 207 in passing. The authors should explain what this is, preferably earlier in the text, as readers that are not well acquainted with PARP biology probably know very little about PARG.

Line 223-225: It is easy to misinterpret the sentence to mean that Chagas disease is a kind of genotoxic stress. Please rephrase to increase clarity.

Line 276: ‘importation’ should be ‘import’

Section 4 outlines BER in nuclei and mitochondria. I am fairly critical of their model in Figure 3, and I think this section provides only a superficial view of the BER pathway in mitochondria. For example, the first two steps are significantly more complex than outlined in Figure 3. There are many bifunctional DNA glycosylases with beta-lyase activity (OGG1, NTHL1), and some even with delta-lyase activity (NEIL1, 2, 3).  This is important, as these enzymatic activities do not generate dRP-moieties at all, which is central to the authors’ further discussion. Secondly, the authors make several internally conflicting statements on the rate-limiting step of BER. At line 292 they claim that dRP removal is the rate-limiting step (supported by two references that are a quarter of a century old), whilst at line 351 they expect that ‘detecting DNA damage’ is the rate-limiting step (with no references). Whilst dRP-removal may be the slowest step biochemically under typical in vitro conditions, the rate of the BER process is also determined by enzyme availability, which in turn is a function of expression levels and competition from other ongoing repair processes.

Furthermore, it is still not definitively proved that BER in mitochondria is exclusively through LP-BER, and even for mitochondrial LP-BER the figure is lacking, since repair of their 6-nt long patch cannot be completed without removal of the displaced strand. Several enzymes capable of flap-removal have been detected in mitochondria, including FEN1, DNA2, and EXOG. Alternatively, if the authors think that single-nucleotide insertion or strand-displacement does not take place during mitochondrial BER, I would love for them to argue their case more strongly. It would yield a more controversial, and threreby more interesting paper. Finally, the BER discussion would be strengthen by also mentioning aprataxin and TDP1 and their involvement in mitochondrial BER.

Section 5 discusses the role of PARP1 in DNA repair. It is overall quite interesting and the writing is very good. Addressing the following might improve the section further:

As mentioned before, their statement on the rate-limiting step of DNA repair could be strengthened significantly by referring to relevant literature investigating this. For example, FRAP studies suggest that at least laser-induced damage is detected is almost immediately in human cells, but resolving the damage is orders of magnitude slower.

At Line 372 the authors mention that a similar PACC as described by the Wilson group has not been observed in mitochondria. However, some groups have reported immunoprecipitation of multi-protein repair complexes from mitochondrial extracts (e.g., PMID: 18635552), which should be mentioned.  Alternatively, the differential organization of mtDNA in membrane-associated nucleoids might make tight, pre-formed complexes superfluous in mitochondria (see PMC5478868 for a review on these structures). This could also be discussed.

Line 398 repeats ‘levels’ three times. Please rephrase to enhance readability.

I find no flaws in the Quality of English Language, which is proabably superior to this reviewers' report.

Author Response

(The authors gave the same response as above.)

Round 2

Reviewer 1 Report

Hermann and Yin provided a revised review of PARP1 in nuclear and mitochondrial BER. Based on the original submission, I was moderately enthusiastic and hoped the authors would respond definitively in their revised version to improve the paper substantially. Unfortunately, I find that the main concerns have not been adequately addressed.

Comment on the rebuttal:

Point 1) The new version still contains inappropriate statements (see other points) and grammatical errors. Furthermore, this reviewer is not trying to debunk what some groups have described about PARP1 in the mitochondria. However, the majority of arguments against mitochondrial localization are neglected by the authors. For example, if one focuses on the localization of PARP1 by immunofluorescence using a specific anti-PARP1 antibody for immunofluorescence, the localization of PARP1 to the mitochondrial can not be observed (https://www.abcam.com/products/primary-antibodies/parp1-antibody-e102-ab32138.html#lb https://www.thermofisher.com/antibody/product/PARP1-Antibody-clone-C-2-10-Monoclonal/MA3-950 https://www.ptglab.com/products/PARP1-Antibody-13371-1-AP.htm https://www.atlasantibodies.com/products/antibodies/primary-antibodies/precisa-monoclonals/parp1-antibody-amab90959/).

Moreover, the most elaborate database for mitochondrial proteins, MitoCarta3.0, does not provide evidence for PARP1 (https://personal.broadinstitute.org/scalvo/MitoCarta3.0/human.mitocarta3.0.html). Also, in other protein databases (Uniprot, Human Protein Atlas), PARP1 is not annotated to the mitochondria.

Point 2) This reviewer did not intend to denounce the publication by Rossi et al., 2009 or Szcznesny et al., 2014. However, Köritzer et al., 2021 (https://doi.org/10.1002/jcb.29887) excluded with a similar technique that PARP1  localizes to the mitochondria. Moreover, should the mitochondrial localization of PARP1 be a reproducible and functionally relevant aspect, it should be observed in different cell lines and conditions (independent of the preparation of samples).

Point 3) The outdating of references refers to references that used “older” technologies to investigate the localization of PARP1. At the same time, newer investigations, including proteomics analyses, did not support the localization of PARP1 in the mitochondria (see argument above).

Point 4) Given this reviewer, the evidence that PARP1 is NOT localizing to the mitochondria is simply more robust and substantiated (even under stress conditions) (https://www.tandfonline.com/doi/full/10.1128/MCB.25.17.7625-7636.2005?scroll=top&needAccess=true&role=tab&aria-labelledby=full-article).

5) The paper still requires proofreading:

Line 9: PARP1 is not a signaling molecule, rewrite: is a critical player in DNA damage repair or similar

Line 32: that a different repair machinery

Line 45: just write electron transport chain or respiratory chain (leave out oxidative phosphorylation)

Line 56: grammar: currently has 17 known members

Line 60: grammar: one has no catalytic

Line 65: once again, PARP1 catalyzes most PARylation only in the context of DNA damage and not at baseline

Line 92: Figure legends are all left-aligned, justify text

Line 214: just for completion add macrodomain containing erasers

Line 217: glyco

Line 308: import

See point 5

Author Response

We thank the reviewer's helpful comments and positive evaluation.

Reviewer 2 Report

Thanks for addressing all my comments, and I wish the authors the best.

Author Response

(The authors gave the same response as above.)
